# Enhancing Object Discovery for Unsupervised Instance Segmentation and Object Detection

## Abstract

We propose **C**ut-**O**nce-and-**LE**a**R**n (COLER), a simple approach for unsupervised instance segmentation and object detection. COLER first uses our developed CutOnce to generate coarse pseudo labels, then enables the detector to learn from these masks. CutOnce applies Normalized Cut (NCut) only once and does not rely on any clustering methods (e.g., K-Means), but it can generate multiple object masks in an image. *Our work opens a new direction for NCut algorithm in multi-object segmentation.* We have designed several novel yet simple modules that not only allow CutOnce to fully leverage the object discovery capabilities of self-supervised model, but also free it from reliance on mask post-processing. During training, COLER achieves strong performance without requiring specially designed loss functions for pseudo labels, and its performance is further improved through self-training. COLER is a zero-shot unsupervised model that outperforms previous state-of-the-art methods on multiple benchmarks. We believe our method can help advance the field of unsupervised object localization.

## 1. Introduction

In computer vision, segmentation tasks heavily rely on large-scale manual annotations, which significantly limits the development speed of the field. To reduce dependence on labeled data, researchers have begun exploring weakly-supervised or unsupervised approaches (Simeoni et al., 2025). This work focuses on exploring how to perform unsupervised instance segmentation and object detection efficiently. Found (Siméoni et al., 2022) established the two-stage framework for unsupervised segmentation: generating pseudo labels followed by training a detector using them.

In recent years, methods that use self-supervised models to extract image features and then apply Normalized Cut (NCut) (Shi & Malik, 2000) to generate pseudo labels have made impressive progress, outperforming many other approaches. TokenCut (Wang et al., 2022b), MaskCut (Wang et al., 2023), and VoteCut (Arica et al., 2024) all use the DINO (Caron et al., 2021) model to extract features and produce reliable pseudo labels, achieving significant advances in unsupervised object localization field. DiffCut (Couairon et al., 2024) employs a diffusion UNet (Ronneberger et al., 2015) encoder for feature extraction, targeting unsupervised semantic segmentation.

The NCut paper (Shi & Malik, 2000) recommends two approaches for extending from single-object to multi-object segmentation. One approach is to perform NCut once, using top eigenvectors followed by clustering (e.g., K-Means). VoteCut (Arica et al., 2024) adopts a clustering-based approach and generates masks using only the second smallest eigenvector. However, *clustering methods require specifying the number of clusters*, which reduces the generality of such approaches. The other approach is to recursively partition the separated groups, that is, to further split the current foreground or background regions. MaskCut (Wang et al., 2023) adopts this strategy by recursively partitioning the background. However, this approach clearly *suffers from error accumulation as the number of recursive steps increases*.

This paper uses NCut to segment multiple objects, but it differs from the two types of methods mentioned above. *Our CutOnce neither relies on multiple applications of NCut nor on clustering methods when discovering multiple objects.* In short, by applying NCut only once, CutOnce can discover multiple objects from an image rather than just one, which is the origin of its name. Furthermore, *CutOnce is capable of detecting masks for over 10 objects (examples are shown in the Appendix), which exceeds the maximum number of detectable targets of currently known methods.* CutOnce does not require computationally expensive methods such as Conditional Random Field (CRF) (Krähenbühl & Koltun, 2011). Table 1 summarizes the key properties of CutOnce and pop-

---

[1]Anonymous Institution, Anonymous City, Anonymous Region, Anonymous Country. Correspondence to: Anonymous Author <anon.email@domain.com>.

Preliminary work. Under review by the International Conference on Machine Learning (ICML). Do not distribute.

*Table 1.* **Key properties of our CutOnce and COLER with state-of-the-art methods.**

| Train-Free Mask Generators | MaskCut | VoteCut | CutOnce |
|---|---|---|---|
| Normalized Cut #nums | 3 | 1 | 1 |
| clustering method | × | ✓ | × |
| post-process mask | ✓ | ✓ | × |
| self-supervised #models | 1 | 6 | 1 |
| max #objects detected | 3 | 10 | >**10** |
| mask generation time (s/img) | 5.6 | 2.4 | **0.23** |

| Pseudo Mask Learners | CutLER | CuVLER | COLER |
|---|---|---|---|
| pseudo mask loss function | ✓ | ✓ | × |
| $AP_{50}^{mask}$ on COCO val2017 | 18.9 | 19.3 | **22.1** |

ular existing methods (Wang et al., 2023; Arica et al., 2024), showing that CutOnce not only detects more objects but also generates annotations up to $10\times$ faster. COLER uses CutOnce's annotations for training and achieves good performance through self-training, without the need to design specialized loss functions to handle errors in the "ground truth" provided by pseudo masks.

The contributions of this paper can be summarized as follows: *1)* We develop an efficient tool, CutOnce, for generating coarse masks. *We introduce a novel paradigm for applying NCut to multi-object segmentation. 2)* We train a detector, COLER, using pseudo masks generated by CutOnce. COLER is a zero-shot model that outperforms prior work across multiple datasets.

## 2. Related Work

**Self-Supervised Vision Transformer.** Self-supervised models are capable of learning deep features without human annotations or supervision. ViT (Dosovitskiy et al., 2020) captures long-range dependencies between different regions in images through a global self-attention mechanism, making it easier to focus on semantically consistent target regions compared to CNN. DINO (Caron et al., 2021) combines both advantages, playing a key role in advancing unsupervised object localization. It adopts a teacher-student training framework and introduces a novel contrastive learning strategy that compares features from the original image and its random crops to learn stronger visual representations. Thanks to the built-in spatial attention mechanism of the ViT architecture, DINO's attention maps can be directly used for localization and have shown advantages over previous methods. By further processing DINO's attention maps, more precise object regions can be obtained. And many methods (Wang et al., 2022b; 2023; Arica et al., 2024; Sick et al., 2025) have been developed based on this idea, achieving significant progress in their respective fields.

**Unsupervised Instance Segmentation and Object Detec-**

tion. The methods introduced in this section can be categorized into the field of unsupervised object localization.

Early studies demonstrated the effectiveness of self-supervised learning, but their performance on large-scale datasets remains unsatisfactory. LOST (Siméoni et al., 2021) is the first to localize objects by leveraging the final layer CLS token from a pre-trained transformer DINO (Caron et al., 2021) and computing patch-wise similarity within single image, but it can only localize one object. MOST (Rambhatla et al., 2023) extends this to multiple objects through entropy-based box analysis and clustering. FreeSOLO (Wang et al., 2022a) uses features from DenseCL (Wang et al., 2020) to generate a set of "queries" and "keys" which are convolved to produce masks and also supports multiple object localization.

TokenCut (Wang et al., 2022b) is the first to apply NCut to features extracted by DINO, significantly improving the quality of pseudo labels, but it can only segment single instance. CutLER (Wang et al., 2023) recursively applies NCut on one image to generate masks for multiple instances. CuVLER (Arica et al., 2024) uses multiple self-supervised models to generate diverse mask proposals and selects the best masks through clustering and pixel-wise voting. In addition, it assigns a confidence score to each pseudo mask. Table 1 presents the properties of the above methods and compares them with ours. CutS3D (Sick et al., 2025) introduces 3D information to enhance segmentation in 2D images, showing certain advantages in handling overlapping or connected objects and demonstrating strong potential for real-world applications. DiffNCut (Liu & Gould, 2024) proposes Differentiable Normalized Cuts. In other words, it uses NCut to propagate gradients and fine-tune DINO. Since NCut enhances object discovery, methods that use it outperform those that do not.

## 3. Method

The overall pipelines of the proposed COLER and CutLER (Wang et al., 2023) are highly similar. Beyond the preliminaries, this chapter provides a detailed description of CutOnce, our efficient mask generator, followed by training and self-training of the detector using pseudo labels.

### 3.1. Preliminaries

**Normalized Cut.** NCut (Shi & Malik, 2000) formulates image segmentation as a graph partitioning problem. It constructs a fully connected undirected graph $\mathbf{G} = (\mathbf{V}, \mathbf{E})$ by representing the image as a set of nodes, where each pair of nodes is connected by an edge with weight $w_{ij}$ indicating their similarity. NCut minimizes the cost of partitioning the graph into two subgraphs by solving a generalized eigen-

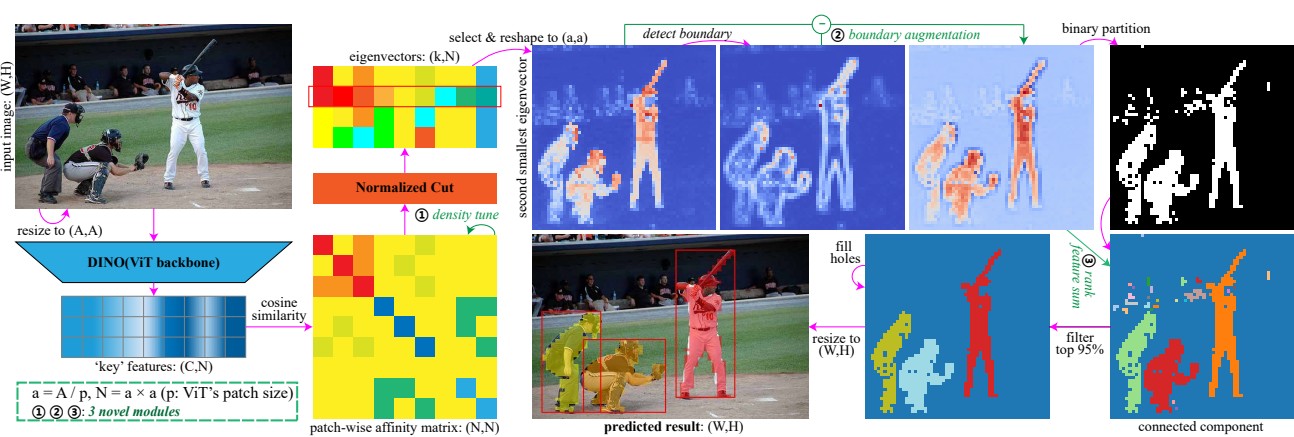

*Figure 1.* **Overview of CutOnce.** First, the resized image is processed by DINO to extract the "key" features. Then, construct the affinity matrix and apply the NCut algorithm to obtain the second smallest eigenvector. Next, the original eigenvector is used to compute the boundary eigenvector, and the two are subtracted element-wise to produce the boundary-enhanced eigenvector. Finally, perform graph partitioning on the enhanced eigenvectors to generate segmentation masks.

value system

$$(\mathbf{D} - \mathbf{W})\mathbf{x} = \lambda \mathbf{D}\mathbf{x} \tag{1}$$

to yield a set of $N \times N$ eigenvectors $\mathbf{x}$, where $N$ denotes the number of nodes. Here, $\mathbf{D}$ is an $N \times N$ diagonal matrix with $d_{ii} = \sum_j w_{ij}$, and $\mathbf{W}$ is an $N \times N$ symmetric matrix representing the adjacency matrix of edge weights.

**TokenCut and MaskCut.** Our CutOnce is *based on the overall workflow of TokenCut* (Wang et al., 2022b), with some implementation details *adopting the design of MaskCut* (Wang et al., 2023). The following describes the consistent parts of CutOnce with existing methods.

First, the input image is passed through a single self-supervised model to extract the "key" features from the last attention layer, denoted as $\mathbf{K} \in \mathbb{R}^{D \times N}$, where $D$ is the feature dimension and $N$ is the number of nodes. The key feature of each patch is represented as a feature vector $\mathbf{k}_i$ $(i = 1, \ldots, N)$. These features encode the spatial information captured by ViT (Dosovitskiy et al., 2020), so the cosine similarity between them can be used to calculate the elements in $\mathbf{W}$.

$$w_{ij} = \cos(\mathbf{k}_i, \mathbf{k}_j) = \frac{\mathbf{k}_i^T \mathbf{k}_j}{\|\mathbf{k}_i\|_2 \, \|\mathbf{k}_j\|_2} \tag{2}$$

Then, the second smallest eigenvector $\mathbf{y}_1$ is obtained from the solution of Equation 1, which can be viewed as *an enhanced attention map*. When the splitting threshold is set to $\overline{\mathbf{y}_1} = \frac{1}{N} \sum_i \mathbf{y}_1^i$, $\mathbf{y}_1$ can be effectively divided into background and foreground. To determine which group corresponds to the foreground, we examine the distribution of $\mathbf{y}_1$ on the ImageNet (Deng et al., 2009) and COCO (Lin et al., 2014) datasets, using criteria similar to MaskCut: *1)* The foreground set contains fewer than three of the four image corners, an idea inspired by the object-centric prior (Maji et al., 2011). *2)* $|\max(\mathbf{y}_1)| > |\min(\mathbf{y}_1)|$. If either of these

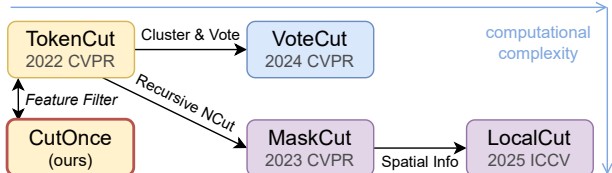

*Figure 2.* **Conceptual comparison of TokenCut-based object discovery methods.** *CutOnce replaces a module in the original TokenCut workflow*, rather than adding new components as in previous methods.

conditions is not satisfied (with condition 1 taking priority), the feature vector $\mathbf{v}$ used for mask partitioning is set to $-\mathbf{y}_1$; otherwise, it is set to $\mathbf{y}_1$. Finally, the groups in $\mathbf{v}$ with values greater than the partition point are regarded as foreground, while those with smaller values are regarded as background.

Existing methods have paved the way for our research, but the *following limitations* remain urgent to address: *1)* Masks are not refined at the NCut stage. *2)* Predicted masks are prone to errors. *3)* Processing multiple targets is relatively time-consuming.

### 3.2. CutOnce: Efficient Mask Generator

Our CutOnce does not recurse NCut, does not rely on clustering, and does not use CRF to refine mask boundaries. As shown in Figure 2, *CutOnce is a variant of TokenCut rather than an incremental extension of existing methods.* For detailed descriptions, please refer to the **Ranking-Based Instance Filter** section. It is, to our knowledge, *the simplest multi-object mask generator to date, with computational complexity only slightly higher than TokenCut.* Given the limitations of prior works, we aim to address the following challenges: *1)* How to produce more accurate foreground boundaries. *2)* How to enable NCut algorithm to discover multiple objects rather than focusing on a single one. *3)*

How to segment more objects without introducing too many incorrect masks.

Figure 1 illustrates the overall pipeline of CutOnce, which consists of **three novel modules**. The first two improvements optimize the *input* and *output* of the NCut algorithm, respectively. They all *make the eigenvector distribution closer to the reality*, thereby addressing the first challenge. The second improvement expands the foreground region, effectively resolving the second challenge. The third improvement is applied in the graph partition stage, effectively filtering out the *most salient multiple objects* and addressing the third challenge. *The theoretical analysis and visualizations of these three modules are provided in the **Appendix**.*

**Density-Tune Cosine Similarity.** Previous methods (Wang et al., 2022b; 2023; Arica et al., 2024) compute the edge weight matrix $\mathbf{W}$ solely based on the cosine similarity between nodes, ignoring the variations in feature density across different image regions. This often leads to *over-activation in certain areas*, which negatively affects boundary localization. To address this issue, we propose a local-density-aware temperature modulation for cosine similarity. The idea is to adaptively adjust the temperature parameter in similarity computation based on the local density of feature points.

For the convenience of subsequent calculations, the deep learning features $\mathbf{K}$ are first normalized. The elements of the adaptive edge weight matrix are defined as:

$$w_{ij} = \frac{\cos(\mathbf{k}_i, \mathbf{k}_j)}{T_{ij}} = \frac{\mathbf{k}_i^T \mathbf{k}_j}{T_0 + \alpha \cdot \frac{\rho_i + \rho_j}{2}} \quad (3)$$

where $T_{ij}$ denotes the adaptive temperature parameter, $T_0$ is the base temperature, $\alpha$ is the modulation parameter, and $\rho_i$ and $\rho_j$ represent the local densities of feature points $i$ and $j$, respectively. The local density is computed by first calculating the pairwise cosine similarity matrix $\mathbf{S} = \mathbf{K}\mathbf{K}^T$ for all patches in batch. Then, for each feature point $i$, we select its top-$k$ most similar neighbors (excluding itself) and compute the local density as:

$$\rho_i = \frac{1}{k} \sum_{j \in \mathcal{N}_k(i)} \mathbf{S}_{ij} \quad (4)$$

where $\mathcal{N}k(i)$ denotes the set of indices corresponding to the $k$ most similar samples to the $i$-th sample (excluding $i$ itself). The modulated $\mathbf{W}$ still requires feature contrast enhancement, following the approach in TokenCut (Wang et al., 2022b). Specifically, $\mathbf{W}_{ij}$ is set to 1 if $\mathbf{W}_{ij} \geq \tau^{\text{ncut}}$, and to $1e^{-5}$ otherwise, where $\tau^{\text{ncut}}$ is set to 0.15 by default.

Background regions typically exhibit relatively uniform feature distributions, and using lower temperatures in these low-density areas preserves the discriminative power of the original similarity. In contrast, object interiors often have dense but uneven features, where abundant redundant local high similarities exist. By introducing higher temperatures to smooth these "overconfident" similarities, we can suppress over-activation and make the similarity distribution within objects more consistent. Obviously, *regions with uniform similarity are more likely to be grouped together.*

This density-tune module shares the same idea as self-tuning spectral clustering (Zelnik-Manor & Perona, 2004), *both optimizing the affinity matrix via neighborhood information to reduce sensitivity*. Besides this, a similar idea is adopted by the clustering algorithm LDP-SC (Long et al., 2022), which combines local density peaks with NCut and demonstrates significant advantages when handling locally tree-structured data. Together, these related approaches provide solid theoretical and empirical justification for our design choice in CutOnce.

**Boundary Augmentation.** *The attention map $\mathbf{y_1}$ output by NCut tends to focus on a single object*, which is the fundamental reason why MaskCut (Wang et al., 2023) applies the NCut algorithm multiple times to discover multiple objects. However, *our goal is to obtain masks for multiple objects using NCut only once*. This raises an important question: Are the less salient objects in the foreground being ignored by the self-supervised model or the NCut algorithm? In fact, the potential objects are already represented, but it is difficult to assign those with relatively low attention to the foreground. From the first attention map (visualization of $\mathbf{y_1}$) in Figure 1, the following information can be easily extracted: *1)* Regions with larger feature values are usually concentrated within parts of the objects. *2)* In some object boundary areas, the feature values differ significantly from their surrounding regions.

Can boundary information be used to encourage a more uniform feature distribution rather than concentrating on a single object? In practice, incorporating boundary information to refine original eigenvector $\mathbf{y_1}$ has proven to be an effective approach. We propose a boundary-enhanced feature representation:

$$X_a = X - X_b \quad (5)$$

where $X$ is the original eigenvector, and $X_b$ is the boundary eigenvector obtained by calculating the difference between each point and its neighborhood:

$$X_b = \frac{1}{k} \sum_{n \in \mathcal{N}_k} |X - X_n|, \quad k \in \{4, 8\} \quad (6)$$

Here, $\mathcal{N}_k$ denotes the $k$-neighborhood, which is set to 8 by default, and $X_n$ represents the feature values within the neighborhood. To correctly compute boundary pixels, padding is applied to the four edges and four corners of the feature map. The padding regions should not introduce new groups, so the original boundary features are simply

extended. Specifically, the feature values in the padding areas are set to those of the adjacent boundary pixels.

The third attention map in Figure 1 demonstrates the benefits of this improvement: *(1) The saliency of more objects is enhanced. (2) Adjacent objects are less likely to be considered as a single entity.* First, we analyze the first advantage. $\mathbf{X}_b$ places high attention on regions with large feature differences, particularly near the boundary between foreground and background as well as certain regions inside the foreground. Within the same foreground target region, positions with larger $\mathbf{X}$ values typically correspond to larger $\mathbf{X}_b$ values. *By smoothing the feature distribution through Equation 5, we can reduce the feature differences between different targets within the foreground*, thus making more objects "noticeable". Next, we analyze the second advantage. $\mathbf{X}_b$ shows high attention on both sides of object boundaries, which causes $\mathbf{X}_a$ to "merge" the areas around the boundary into the background, *making all detected objects smaller*. For objects that are close to each other, the gaps between them are enlarged, which facilitates their correct separation. However, for small objects, prediction errors increase. If an object is very small in area, this may cause such "noisy points" to vanish. Considering both comprehensive theoretical analysis and subsequent ablation study, boundary enhancement proves to be a strategy with more advantages than disadvantages.

The design of the boundary enhancement module is inspired by the residual connections in ResNet (He et al., 2016), where an auxiliary path is used to correct the main path. Our $\mathbf{X}_b$ is a local difference (a first-order gradient approximation) computed from the eigenvector, *capturing boundary-related variation patterns*. The idea is similar to the classical Laplacian of Gaussian (LoG) (Marr & Hildreth, 1980) in image processing, *where strong local changes (the sum of second-order derivatives) are used for edge detection*.

**Ranking-Based Instance Filter.** After extracting the foreground region from eigenvector using a segmentation threshold (as described in the *Preliminaries* section), the next step is to separate multiple objects from the foreground. To achieve this, we first apply 4-connectivity to perform connected component decomposition on the foreground region, treating each connected component as a candidate object. To select multiple salient objects from these candidates, we propose a feature rank-based connected component filtering strategy, which proceeds as follows:

1. Sort: Suppose there are $N$ candidate object regions. Let $s_i$ denote the feature sum of the $i$-th region. Sort all feature sums $\{s_i\}_{i=1}^N$ in descending order to obtain the index sequence $\{i_1, i_2, \ldots, i_N\}$.
2. Cumulative screening: Select the top-ranked objects one by one until the cumulative feature proportion reaches:

$\frac{\sum_{j=1}^k s_{i_j}}{\sum_{i=1}^N s_i} \geq \tau$, where $\tau \in (0, 1)$ is the feature preservation threshold.

3. Output: The masks corresponding to the top $k$ selected objects.

Previous methods (TokenCut and CutLER) determine the sole target by checking the connected component containing the maximum absolute value in the eigenvector. This point-based criterion is prone to misidentification. *We instead use the features of a region rather than a single point to decide which object NCut prioritizes.* This approach accounts for both the spatial extent of an object and the feature magnitude at individual points, allowing us to select the most salient objects and output them in descending order. More importantly, it introduces only a single hyperparameter.

### 3.3. Detector Training and Self-Training

The training details of COLER are similar to those of CutLER, *with the main difference being that no specific loss function is used*. After training the detector with pseudo labels generated by CutOnce, the detector can identify more masks than those in the pseudo labels. Therefore, we adopt a self-training strategy to further improve the model performance. In the $t$-th round of self-training ($t \in 1, 2, \ldots$), we first perform inference on the training data using the current model, retaining predicted masks with confidence scores higher than $0.6 - 0.05t$ as high-quality pseudo-labels. To avoid label duplication and maintain diversity in the training data, we also select a subset of pseudo-labels from round $(t-1)$ whose IoU with the current high-confidence predictions is $< 0.5$. The final training labels for round $t$ are obtained by merging these two sets.

## 4. Experiments

### 4.1. Implementation Details

**Datasets.** In this paper, only the images from the train split of ImageNet-1K (Deng et al., 2009) (1.28M images) are used for all training processes of the COLER model, with no manual annotations or any supervised pre-trained models employed in the training. *Due to limited computational resources, we use the ImageNet val split (50K images) to generate pseudo-labels in our ablation study, while keeping all other settings identical.* This variant is denoted as CutOnce* and COLER*.

We evaluate on two subsets of the COCO (Lin et al., 2014) dataset, LVIS (Gupta et al., 2019), VOC (Everingham et al., 2010), KITTI (Geiger et al., 2012), OpenImages (Kuznetsova et al., 2020) and Objects365 (Shao et al., 2019), resulting in a total of 7 benchmarks. This paper mainly uses $\text{AP}_{50}$ and AP as the evaluation metrics for presenting results.

*Table 2.* **Evaluation of pseudo labels.** #N denotes the average number of masks per image.

| Datasets → Methods | Use CRF | ImageNet val | | | | | COCO val2017 | | | | | | | | |
|---|---|---|---|---|---|---|---|---|---|---|---|---|---|---|---|
| | | $AP^{box}$ | $AP^{box}_{50}$ | $AP^{box}_{75}$ | $AR^{box}_{100}$ | #N | $AP^{box}$ | $AP^{box}_{50}$ | $AP^{box}_{75}$ | $AR^{box}_{100}$ | $AP^{mask}$ | $AP^{mask}_{50}$ | $AP^{mask}_{75}$ | $AR^{mask}_{100}$ | #N |
| MaskCut | ✓ | 10.6 | 20.3 | 10.0 | 27.7 | 1.9 | 3.9 | 7.9 | 3.3 | 7.7 | 3.1 | 6.8 | 2.5 | 6.5 | 1.9 |
| VoteCut | ✓ | **20.9** | **36.2** | **20.0** | **45.0** | 8.9 | **5.5** | **10.7** | **4.9** | **12.2** | **4.5** | **9.3** | **3.9** | **10.3** | 8.6 |
| CutOnce(ours) | × | 16.5 | 32.5 | 15.0 | 31.5 | 1.8 | 4.1 | 8.2 | 3.6 | 7.6 | 3.1 | 7.0 | 2.4 | 6.0 | 1.8 |
| CutOnce+(ours) | ✓ | 16.9 | 32.6 | 15.4 | 32.3 | 1.8 | 4.2 | 8.2 | 3.7 | 7.9 | 3.4 | 7.2 | 2.9 | 6.8 | 1.8 |

**CutOnce.** We resize images to $480 \times 480$ pixels and use the ViT-B/8 (Dosovitskiy et al., 2020) DINO (Caron et al., 2021) model by default to extract features. For the density-tune similarity module, $k$, $T_0$, and $\alpha$ are set to 10, 1.0, and 0.5, respectively. The filter parameter $\tau$ is set to 0.95.

**CAD (class-agnostic detector).** *All training and inference are conducted on single NVIDIA RTX 4090 GPU.* All experiments are implemented on the detectron2 (Wu et al., 2019) platform using Cascade Mask R-CNN (Cai & Vasconcelos, 2017) as the default detector. The detector is trained with masks and bounding boxes generated by CutOnce for 80K iterations with copy-paste augmentation (Ghiasi et al., 2021). The batch size is set to 8, learning rate to 0.01, weight decay to $5 \times 10^{-5}$, and momentum to 0.9. Following the same setup as the copy-paste augmentation in CutLER (Wang et al., 2023), we randomly downsample the masks with a scale factor uniformly sampled from 0.3 to 1.0.

**Self-Training.** In this stage, the model is initialized with the weights from the previous phase and trained for 60K iterations. Other settings follow CutLER, with the learning rate set to 0.005 and the copy-paste augmentation scalar uniformly sampled between 0.5 and 1.0.

**State-of-the-art (SOTA) Comparison.** We compare our method with CutLER (Wang et al., 2023) and CuVLER (Arica et al., 2024). CuVLER provides two versions of pretrained weights: *zero-shot and COCO self-train, and we only use the former. Some methods are excluded from comparison.* For example, CutS3D (Sick et al., 2025) has not released its source code and weights, and unMORE (Yang et al., 2025) leverages the COCO dataset for training, which is incompatible with zero-shot evaluation.

### 4.2. Pseudo Labels Evaluation

To evaluate the pseudo masks using the official COCO (Lin et al., 2014) evaluation tool, each annotation must be assigned a confidence score. *The score has a significant impact on AP but does not affect AR.* Since VoteCut (Arica et al., 2024) comes with its own scoring mechanism, we retain its original setting. To ensure a relatively fair comparison, we apply the same scoring scheme to both Mask-Cut (Wang et al., 2023) and CutOnce. The reason is that both methods output masks in descending order of object saliency and treat all outputs as "ground truth". For multiple

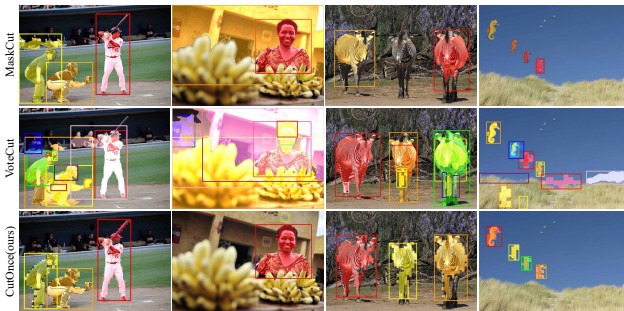

*Figure 3.* **Qualitative comparison between our CutOnce and related methods on COCO val2017.** All methods display all predicted masks.

masks associated with the same image, we adopt a linearly decreasing score assignment scheme. The confidence score for each mask is defined as 1.0 if $N = 1$, and as $1.0 - \frac{k}{2N-2}$ if $N > 1$, where $N$ is the total number of masks in the image and $k$ is the index of the current mask ($k = 0, 1, \ldots, N-1$). This ensures that the first mask always receives the highest confidence score of 1.0, while the last mask is assigned a score of 0.5.

Table 2 presents the quantitative comparisons of different methods, and Figure 3 shows the corresponding qualitative results. On ImageNet val, all metrics of CutOnce lie between those of MaskCut and VoteCut. On COCO val2017, CutOnce shows poor performance in $AR^{mask}_{100}$. From the performance of *CutOnce+ (The only difference from CutOnce is whether CRF is used or not)*, it clearly outperforms CutOnce and MaskCut, achieving substantial improvements in all metrics except $AP_{50}$. Evidently, using CRF to refine the masks has a positive effect on high-precision localization, while having little impact on coarse localization, ultimately leading to significant gains in AP and AR. As shown in Figure 3, the targets localized by CutOnce are always semantically correct. *Since CutOnce does not use CRF for post-processing, its boundaries are slightly coarse compared to other methods.* MaskCut applies NCut three times and can localize at most three objects. *This design clearly cannot adapt to scenarios with either many objects or very few.* VoteCut uses clustering and integrates outputs from multiple self-supervised models, producing a large number of masks, some correct and some not meaningful. Although VoteCut appears to achieve the highest metrics, *its incorrect masks can negatively impact training.* Overall, the pseudo

*Table 3.* **Zero-shot evaluation across three COCO-based datasets.** IN and '1 + 3' denote ImageNet-1K and one training plus three rounds of in-domain self-training, respectively.

| Datasets → Methods | Pretrain | Train #Rounds | COCO 20K | | | | COCO val2017 | | | | LVIS | | | |
|---|---|---|---|---|---|---|---|---|---|---|---|---|---|---|
| | | | $AP^{box}$ | $AP_{50}^{box}$ | $AP^{mask}$ | $AP_{50}^{mask}$ | $AP^{box}$ | $AP_{50}^{box}$ | $AP^{mask}$ | $AP_{50}^{mask}$ | $AP^{box}$ | $AP_{50}^{box}$ | $AP^{mask}$ | $AP_{50}^{mask}$ |
| CutLER | IN train | 1 + 3 | 12.5 | 22.4 | 10.0 | 19.6 | 12.3 | 21.9 | 9.7 | 18.9 | 4.5 | 8.4 | 3.5 | 6.7 |
| CuVLER | IN val | 1 | 12.7 | 23.5 | 10.0 | 20.1 | 12.6 | 23.0 | 9.8 | 19.3 | 4.5 | 8.6 | 3.6 | 6.9 |
| COLER(ours) | IN train | 1 + 3 | 13.3 | 25.2 | 10.8 | 22.3 | 13.1 | 24.9 | 10.5 | 22.1 | 5.0 | 9.6 | 4.0 | 8.1 |
| COLER*(ours) | IN val | 1 + 1 | 12.6 | 24.1 | 9.8 | 20.5 | 12.5 | 23.8 | 9.6 | 20.1 | 4.6 | 9.2 | 3.7 | 7.3 |
| *vs. SOTA (%)* | | | 4.7 | 7.2 | 8.0 | 10.9 | 4.0 | 8.3 | 7.1 | 14.5 | 11.1 | 11.6 | 11.1 | 17.4 |

*Table 4.* **Zero-shot unsupervised object detection evaluation.** Avg. denotes the average value.

| Datasets → Metrics → | Avg. | | COCO | | COCO20K | | LVIS | | VOC | | KITTI | | OpenImages | | Objects365 | |
|---|---|---|---|---|---|---|---|---|---|---|---|---|---|---|---|---|
| | $AP_{50}$ | AP | $AP_{50}$ | AP | $AP_{50}$ | AP | $AP_{50}$ | AP | $AP_{50}$ | AP | $AP_{50}$ | AP | $AP_{50}$ | AP | $AP_{50}$ | AP |
| CutLER | 21.0 | 11.3 | 21.9 | 12.3 | 22.4 | 12.5 | 8.4 | 4.5 | 36.9 | 20.2 | 18.4 | 8.5 | 17.3 | 9.7 | 21.6 | 11.4 |
| CuVLER | 21.2 | 11.4 | 23.0 | 12.6 | 23.5 | 12.7 | 8.6 | 4.5 | 39.4 | 22.3 | 13.0 | 5.1 | 19.6 | 11.6 | 21.6 | 10.9 |
| COLER(ours) | 23.7 | 12.1 | 24.9 | 13.1 | 25.2 | 13.3 | 9.6 | 5.0 | 41.8 | 21.6 | 22.3 | 9.6 | 18.2 | 10.1 | 23.9 | 12.0 |
| COLER*(ours) | 22.3 | 11.4 | 23.8 | 12.5 | 24.1 | 12.6 | 9.2 | 4.6 | 39.1 | 20.5 | 20.8 | 8.8 | 16.7 | 9.3 | 22.6 | 11.2 |
| *vs. SOTA (%)* | 11.6 | 6.3 | 8.3 | 4.0 | 7.2 | 4.7 | 11.6 | 11.1 | 6.1 | -3.1 | 21.2 | 12.9 | -7.1 | -12.9 | 10.6 | 5.3 |

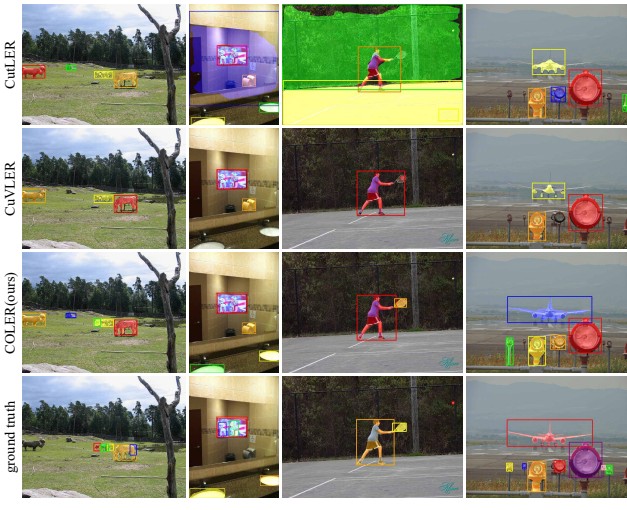

*Figure 4.* **Qualitative comparison between our COLER and SOTA methods on COCO val2017.** Only predictions with confidence ≥ 0.5 are shown.

labels generated by CutOnce contain *relatively fewer noisy labels, which is beneficial for model learning*.

### 4.3. Unsupervised Zero-shot Evaluations

We evaluate COLER on *7 different benchmarks*, containing a variety of object categories and image styles, to validate its effectiveness as a general unsupervised method.

**Detailed in COCO Datasets.** COCO 20K, COCO val2017, and LVIS are all from COCO and provide segmentation annotations, with the corresponding results reported in Table 3. Our COLER achieves leading performance on all three datasets, and COLER* performs comparably to prior methods. The number of self-training iterations reported in Table 3 corresponds to the *maximum iterations that continue to improve performance*. Both CutLER and our COLER stop improving after three rounds of self-training. *CuVLER does not provide in-domain self-trained weights, and its source code does not get weights better than the zero-shot version*. Our COLER* stops improving after the second round of self-training. In terms of percentage improvement over SOTA, *COLER achieves larger gains on the densely annotated LVIS dataset*. Figure 4 shows the qualitative results of COLER compared with related methods. Obviously, *COLER often detects more useful instances, including some that are not annotated in the ground truth*.

**Object Detection.** In Table 4, we report COLER's object detection performance across all datasets. On average, COLER shows a larger improvement in AP50 but a smaller gain in AP. This is expected, as the absence of CRF makes high-precision localization less evident than the improvement seen in AP50. Comparing the best results on each dataset, COLER shows significant advantages on KITTI and LVIS, while performing the worst on OpenImages. The prediction performance of COLER* is also acceptable, which is consistent with the experimental results in the previous section. Overall, our method achieves certain advantages across all datasets.

### 4.4. Ablation Study

Figure 5 visualizes the ablation study of two NCut refinement modules introduced in CutOnce for enhanced object

*Table 5.* **Ablation study of COLER\* hyperparameters on COCO val2017 .** $\tau$ denotes the preservation ratio in the filter, while $k$, $T_0$, and $\alpha$ are parameters related to the adaptive edge weight matrix.

| $\tau$ | 0.8 | 0.9 | 0.95 | 0.99 | $k$ | 3 | 5 | 10 | 20 | $T_0$ | 0.8 | 1.0 | 1.2 | $\alpha$ | 0.3 | 0.5 | 0.7 |
|---|---|---|---|---|---|---|---|---|---|---|---|---|---|---|---|---|---|
| $AP_{50}^{mask}$ | 17.7 | 18.9 | 19.6 | 17.5 | $AP_{50}^{mask}$ | 18.9 | 19.3 | 19.6 | 18.7 | $AP_{50}^{mask}$ | 18.5 | 19.6 | 18.9 | $AP_{50}^{mask}$ | 18.7 | 19.6 | 19.0 |

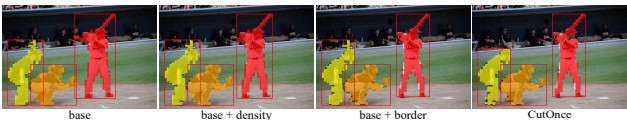

*Figure 5.* **Ablation study on the two NCut refinement modules in CutOnce.**

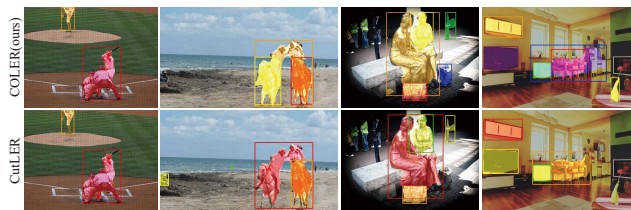

*Figure 6.* **Failure cases of COLER on COCO val2017.**

discovery. It is evident that when the two modules are combined, CutOnce can finely segment the three most salient targets, whereas removing either module often results in poor mask boundaries for certain objects.

Table 5 reports hyperparameter ablation for COLER\* *with one-time detector training*.

Table 6 reports the impact of each component in COLER on the final results across two datasets. Each component contributes to performance improvements to varying degrees, but *boundary augmentation module proves to be the most critical for boosting performance*. Additionally, copy-paste augmentation (Ghiasi et al., 2021) and self-training during the training process also contribute significantly to the overall improvement of COLER.

Table 7 reports the impact of the number of self-training rounds on the final results of COLER. The results after the first, second, and third rounds of self-training show steady improvements, while the gain in the fourth round becomes almost negligible. *By default, COLER uses 3 rounds of self-training*.

### 4.5. Limitations and Future Work

Despite the strong performance of COLER, it also has several limitations. Examples of failure cases are shown in Figure 6: *1)* For heavily overlapping objects with unclear boundaries, the ability to distinguish them correctly is limited. *2)* The number of detected objects is also constrained; although CutOnce can detect more than ten objects, it ultimately depends on the object discovery capability of the self-supervised model. *3)* Its understanding of dense scenes

*Table 6.* **Ablation analysis of COLER components on COCO val2017 and KITTI.**

| Methods | COCO | | KITTI | |
|---|---|---|---|---|
| | $AP^{mask}$ | $AP_{50}^{mask}$ | $AP^{box}$ | $AP_{50}^{box}$ |
| TokenCut + CAD | 6.2 | 14 | 5.7 | 14.1 |
| + rank feature filter | 8.1 | 16.1 | 6.7 | 15.8 |
| + similarity tune | 8.4 | 16.5 | 7.1 | 16.4 |
| + boundary augment | 9.1 | 19.1 | 8.2 | 19.2 |
| + copy-paste | 9.6 | 20.6 | 8.6 | 20.1 |
| + self-training (COLER) | 10.5 | 22.1 | 9.6 | 22.3 |

*Table 7.* **Number of self-training rounds in COLER.**

| Round | COCO | | | | KITTI | |
|---|---|---|---|---|---|---|
| | $AP^{box}$ | $AP_{50}^{box}$ | $AP^{mask}$ | $AP_{50}^{mask}$ | $AP^{box}$ | $AP_{50}^{box}$ |
| 0 | 12.6 | 23.9 | 9.6 | 20.1 | 8.5 | 19.9 |
| 1 | 12.7 | 24.2 | 9.9 | 20.7 | 8.8 | 20.8 |
| 2 | 13.0 | 24.7 | 10.3 | 21.6 | 9.4 | 21.8 |
| 3 | 13.1 | 24.9 | 10.5 | 22.1 | 9.6 | 22.3 |
| 4 | 13.1 | 24.9 | 10.5 | 22.3 | 9.6 | 22.4 |

remains weak, even though it shows notable improvements over previous methods.

The contributions of this work are mainly focused on the pseudo-mask generation stage, while no new mechanisms are introduced in the model training phase. Recent advances in unsupervised panoptic segmentation (Hahn et al., 2025) adopt a teacher–student framework for pseudo-label training, which leads to only marginal improvements. Therefore, how to incorporate more effective training strategies remains an important direction for unsupervised segmentation.

## 5. Conclusion

We propose CutOnce, a novel training-free method for unsupervised object discovery that efficiently and accurately partitions multiple instances in a single image. In particular, the *boundary augmentation strategy stands out as the simplest yet most effective improvement* in this work, and we believe it holds great potential for broader applications in the future. We also introduce COLER, a zero-shot model trained using masks generated by CutOnce. With only ImageNet-1K as the source domain, COLER surpasses previous state-of-the-art models across multiple benchmarks in both unsupervised instance segmentation and object detection.

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

# A. Appendix

The appendix is organized into three main parts:

## A.1. Theoretical Analysis

**Density-Tune Cosine Similarity.** Why do high-density regions require higher temperatures? We can understand this design from the perspective of information redundancy. In feature-dense regions (e.g., texture-rich areas within objects), neighboring feature points are often highly similar. This similarity partially stems from genuine semantic associations but also contains substantial redundant information. If these high similarity values are used directly, the graph cut algorithm will overly trust these local patterns, leading to over-activation problems.

By introducing higher temperature parameters, we are essentially smoothing these "overconfident" similarities. This is similar to using temperature parameters in statistics to control the peakedness of probability distributions—high temperature flattens the distribution, while low temperature sharpens it. In our scenario:

- **Low-density regions (e.g., background)**: Features are sparse with clear differences, and the similarities themselves have good discriminative power, so low temperature preserves the original similarities.
- **High-density regions (e.g., object interiors)**: Features are dense and highly correlated, so high temperature reduces similarity peaks, making similarities within the region more uniform.

This adaptive adjustment mechanism aligns with the core idea of self-tuning spectral clustering (Zelnik-Manor & Perona, 2004): utilizing local neighborhood information to calibrate the global similarity matrix, thereby reducing the algorithm's sensitivity to local density variations.

**Boundary Augmentation.** Figure 7 shows a visualization of the intermediate computation process of *CutOnce's boundary enhancement module*. Obviously, the mechanism of this module is easy to understand and shows immediate effectiveness. What is the mathematical essence of boundary augmentation? Our boundary augmentation operation $X_a = X - X_b$ can be understood from the perspective of *gradient penalization*. In image processing and variational methods, a classical idea is to achieve edge-preserving

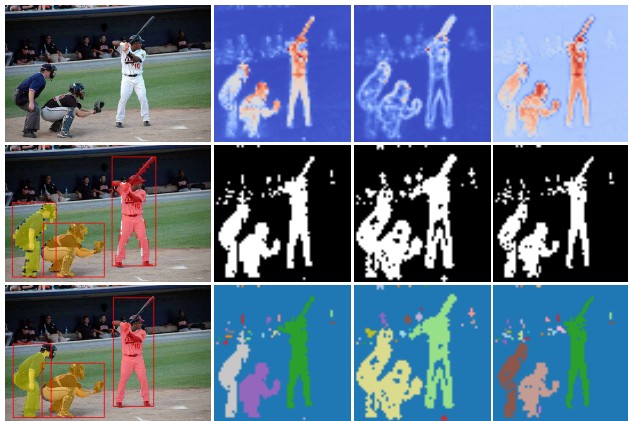

*Figure 7.* **Visualization of the computation process of CutOnce.** The first column (top to bottom) shows the original image, CutOnce prediction, and CutOnce with post-processing. The second to fourth columns show *raw eigenvector*, *boundary eigenvector*, *difference between the two*, and the corresponding foreground-background *binary maps* and *connected component maps*.

smoothing by penalizing gradients. Specifically:

- $X_b$ computes the local differences of the feature map, which is essentially a discrete approximation of the first-order gradient:

$$X_b \approx |\nabla X| \tag{7}$$

- Subtracting $X_b$ is equivalent to applying gradient penalization to the original features, producing two effects:
  - **Intra-region smoothing**: In regions with gradual feature changes (object interiors), $X_b$ values are small, so $X_a$ remains close to the original values.
  - **Boundary suppression**: In regions with drastic feature changes (object boundaries), $X_b$ values are large, significantly reducing $X_a$.

This design shares similarities with the classical *Mumford-Shah image segmentation model* (Mumford & Shah, 1989). The energy function of this model includes two terms:

- Intra-region homogeneity (encouraging feature consistency within the same region)
- Total boundary length (penalizing excessive boundaries)

Our method achieves similar effects through $X - X_b$: reducing the saliency of boundary regions, making it easier for the NCut algorithm to group regions with consistent internal features together. This explains why boundary augmentation can:

- Expand foreground salient regions (smoother interiors)
- Separate adjacent objects (enlarged boundary gaps)

## A.2. Segmenting More Than 10 Objects

**Core mechanism: Adaptive cumulative feature proportion.** Our Ranking-Based Instance Filter uses cumulative

*Figure 8.* **Detecting many objects with CutOnce**. The first row shows the results of CutOnce, and the second row presents the results of CutOnce with post-processing. Both methods successfully detect *17 objects*.

feature proportion as the selection criterion, with a default threshold of 95%. The algorithm continuously selects top-ranked connected regions until their cumulative feature sum reaches 95% of the total foreground features.

The key advantage of this design is *automatic adaptation to the number of objects*: In complex scenes with multiple objects, if there are 10 objects with similar saliency, the top few objects may only account for 30-40% of the total features, so the algorithm naturally continues selecting subsequent objects until reaching 95%. In contrast, if a scene contains only 1-2 dominant objects that occupy over 95% of the features, the algorithm automatically stops.

**Segment 10+ objects.** Figure 8 demonstrates the strong capability of CutOnce in segmenting multiple objects, which previous methods were unable to detect in such quantity. In the scene shown in Figure 8, there are numerous objects with similar sizes, resulting in a relatively uniform saliency distribution. The top-ranked object may only account for about 5% of the total features, the top 10 accumulate to approximately 50-60%, thus requiring selection up to the 15-20th object to reach the 95% threshold. This is precisely why our method can successfully segment 10+ objects in a single NCut operation. It is worth noting that due to the high resolution of the image, after resizing, each patch occupies a larger proportion of the object region, resulting in relatively coarse boundaries. Therefore, CRF post-processing can achieve smoother boundaries.

### A.3. Datasets Used

**COCO** (Lin et al., 2014) (Microsoft Common Objects in Context) is a large-scale dataset for object detection and segmentation. In this paper, COCO refers to the 5k images from the `2017 validation` set.

**COCO 20K**(Lin et al., 2014) contains 19,817 images, a subset of COCO train2014. Many previous unsupervised methods(Wang et al., 2022b; 2023; Arica et al., 2024) have used this dataset to evaluate model performance.

*Table 8.* **Summary of datasets used for zero-shot evaluation (except ImageNet).** "avg. # obj." denotes the average number of annotations per image.

| datasets | testing data | seg label | #images | avg. # obj. |
|---|---|---|---|---|
| COCO | val2017 | ✓ | 5,000 | 7.4 |
| COCO20K | train2014 | ✓ | 19,817 | 7.3 |
| LVIS | val | ✓ | 19,809 | 12.4 |
| Pascal VOC | trainval07 | ✗ | 9,963 | 3.1 |
| KITTI | trainval | ✗ | 7,521 | 4.7 |
| OpenImages V7 | val | ✗ | 41,620 | 7.3 |
| Object365 V2 | val | ✗ | 80,000 | 15.5 |
| ImageNet | val | ✗ | 50,000 | 1.6 |

**LVIS** (Gupta et al., 2019): (Large Vocabulary Instance Segmentation) is a dataset for long-tail instance segmentation. It contains 2.2 million high-quality instance masks of over 1,000 entry-level object categories, collected based on the COCO dataset. In this paper, LVIS refers to the 19,809 images in the `validation` set.

**VOC** (Everingham et al., 2010) (PASCAL Visual Object Classes) is a widely used benchmark for object detection. We evaluate on its `trainval07` split.

**KITTI** (Geiger et al., 2012) (Karlsruhe Institute of Technology and Toyota Technological Institute) is one of the most popular datasets for mobile robotics and autonomous driving. We evaluate on its `trainval` split.

**OpenImages V7** (Kuznetsova et al., 2020) contains multiple tasks, including image classification, object detection, instance segmentation, and visual relationship detection. We evaluate on over 40K images from the `val` split.

**Object365 V2** (Shao et al., 2019) provides a supervised object detection benchmark with a focus on diverse objects in the natural world. We evaluate on 80K images from the `val` split.

The summary of these datasets used for zero-shot evaluation is provided in Table 8.

*Table 9.* **Unsupervised instance segmentation results on all benchmarks in this work.**

| Datasets | $AP^{mask}$ | $AP_{50}^{mask}$ | $AP_{75}^{mask}$ | $AP_S^{mask}$ | $AP_M^{mask}$ | $AP_L^{mask}$ | $AR_1^{mask}$ | $AR_{10}^{mask}$ | $AR_{100}^{mask}$ | $AR_S^{mask}$ | $AR_M^{mask}$ | $AR_L^{mask}$ |
|---|---|---|---|---|---|---|---|---|---|---|---|---|
| COCO | 10.5 | 22.1 | 9.4 | 2.7 | 11.3 | 22.8 | 5.8 | 16.7 | 26.0 | 9.8 | 31.6 | 45.3 |
| COCO20K | 10.8 | 22.3 | 9.6 | 2.9 | 11.4 | 23.0 | 5.9 | 16.9 | 26.3 | 10.0 | 32.0 | 45.4 |
| LVIS | 4.0 | 8.1 | 3.6 | 1.7 | 7.4 | 12.8 | 2.2 | 8.2 | 16.6 | 6.6 | 29.6 | 42.2 |

*Table 10.* **Unsupervised object detection results on all benchmarks in this work.**

| Datasets | $AP^{box}$ | $AP_{50}^{box}$ | $AP_{75}^{box}$ | $AP_S^{box}$ | $AP_M^{box}$ | $AP_L^{box}$ | $AR_1^{box}$ | $AR_{10}^{box}$ | $AR_{100}^{box}$ | $AR_S^{box}$ | $AR_M^{box}$ | $AR_L^{box}$ |
|---|---|---|---|---|---|---|---|---|---|---|---|---|
| COCO | 13.1 | 24.9 | 12.5 | 4.4 | 14.2 | 28.6 | 6.7 | 20.1 | 32.5 | 13.4 | 39.1 | 56.2 |
| COCO20K | 13.3 | 25.2 | 12.6 | 4.7 | 14.1 | 28.8 | 6.8 | 20.4 | 32.8 | 13.7 | 39.4 | 56.3 |
| LVIS | 5.0 | 9.6 | 4.5 | 2.6 | 9.2 | 16.3 | 2.5 | 9.8 | 20.5 | 8.9 | 35.2 | 52.0 |
| VOC | 21.6 | 41.8 | 20.8 | 3.1 | 8.9 | 32.8 | 16.2 | 33.6 | 45.1 | 19.7 | 36.9 | 54.9 |
| KITTI | 9.6 | 22.3 | 6.9 | 1.4 | 6.8 | 18.5 | 6.5 | 20.1 | 30.5 | 17.7 | 27.2 | 43.6 |
| OpenImages | 10.1 | 18.2 | 9.8 | 0.4 | 2.4 | 15.4 | 6.8 | 16.8 | 27.8 | 4.4 | 20.1 | 35.4 |
| Objects365 | 12.0 | 23.9 | 10.6 | 2.9 | 11.1 | 20.2 | 3.0 | 15.4 | 32.5 | 12.0 | 35.0 | 46.4 |

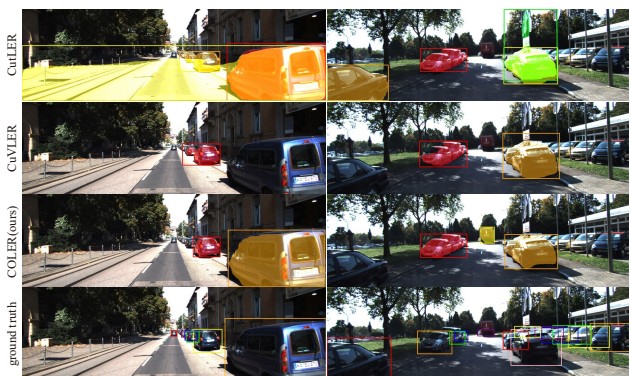

*Figure 9.* **Qualitative comparison of our COLER with previous SOTA methods on KITTI.**

## A.4. Detailed COLER Results

Table 9 and Table 10 present the zero-shot evaluation results on unsupervised instance segmentation and object detection tasks across various datasets, respectively.

## A.5. Other Visualizations

Figure 9 and Figure 10 show additional visualization results of our COLER method compared to previous state-of-the-art approaches. These figures only display predicted results with a confidence score of *no less than 0.5* (ground truth is excluded).

*Figure 10.* **Qualitative comparison of our COLER previous SOTA methods on LVIS, VOC, OpenImages, and Objects365.**

