# OpenReview forum: "Enhancing Object Discovery for Unsupervised Instance Segmentation and Object Detection"
_ICML.cc/2026/Conference — Submitted to ICML 2026_

### Official Review · Reviewer_NtDB · 2026-03-05

**Soundness:** 2
**Presentation:** 3
**Significance:** 2
**Originality:** 2
**Overall Recommendation:** 3
**Confidence:** 5

**Summary:**

This paper proposes the COLER framework for unsupervised instance segmentation and object detection. Its central contribution is CutOnce — a training-free module that performs a single Normalized Cut to simultaneously segment multiple object instances. The method is evaluated on seven benchmarks including COCO, LVIS, VOC, KITTI, OpenImages, and Objects365.

**Compliance With Llm Reviewing Policy:**

Affirmed.

**Final Justification:**

The rebuttal addressed my questions with additional experiments, though the comparison with CutLER variants remains not fully controlled (different losses, datasets, and training settings), which limits the strength of the empirical conclusions. The gain from boundary augmentation on OpenImages is also relatively marginal. I raise my score to 3, acknowledging the authors' effort in the rebuttal.

**Key Questions For Authors:**

## Key Questions For Authors

**Q1.**
CutOnce pseudo-labels underperform VoteCut across all metrics (~1.8 masks/image). The claimed advantage of "fewer noisy labels" lacks quantitative evidence. A controlled comparison training the same detector with CutOnce vs. VoteCut labels under identical settings is needed to substantiate this argument.

**Q2.**
COLER's AP declines on OpenImages (dense scenes, small objects). Is this due to adverse effects of Boundary Augmentation? How does OpenImages performance change when it is ablated? Does the "outperforms SOTA" claim in the abstract warrant qualification? An error analysis would help clarify generalization boundaries in dense settings.

**Limitations:**

Please see Weaknesses.

**Strengths And Weaknesses:**

### Strengths

1. COLER simplifies pseudo-label generation for multi-object instance segmentation into a single NCut operation plus post-processing. Compared to recursive NCut and multi-model ensemble approaches, this yields a 10–24x speedup, demonstrating substantial practical value.

2. Boundary Augmentation provides the largest performance gain in the ablation study (AP^mask_50: 16.5 → 19.1), and the ablation analysis is thorough and transparent.

### Weaknesses

**W1.** COLER uses ~26x more training data than CuVLER (full ImageNet ~1.28M images over 4 rounds vs. 50K images, single round). Under data-equivalent comparison (COLER*), the margin is modest, suggesting limited methodological gains. Comparison with CutLER at the same scale does confirm contributions beyond data scaling, but a clearer disentanglement of method vs. data-scale effects is needed. Baseline coverage is also insufficient — methods like DiffNCut (ECCV 2024) from Related Work lack quantitative comparison.

**W2.** All three modules adapt existing techniques (Density-tune from self-tuning spectral clustering; Boundary Augmentation from Unsharp Masking with reversed objective; Ranking Filter from energy-based thresholding). The pipeline heavily reuses CutLER (acknowledged as "highly similar"), with no training-stage novelty. The modules lack theoretical connections and are better characterized as independent engineering improvements.

---

> ### Author Rebuttal · Authors · 2026-03-26
>
> We thank the reviewer for the detailed discussion on experimental aspects. We will carefully address your questions, and we also welcome more constructive feedback to further improve this work.
>
> Our paper focused on presenting the details of our method, its differences from existing approaches, and the reasoning process behind it. The advantage is that most reviewers found the paper easy to understand; however, some details were not sufficiently clarified.
>
> ***Our CutOnce introduces a third paradigm for multi-object segmentation based on NCut. If similar work exists, we kindly ask the reviewer to explicitly point it out.*** Unlike other methods, all engineering details are presented directly in the main paper.
>
> Recently, we observed that **COLER** achieves significantly larger improvements on scene-centric datasets such as Cityscapes, which is consistent with our findings on KITTI. **This highlights the strong object localization capability of COLER on scene-centric datasets.**
> | Method | $AP_{mask}$ | $AP^{50}_{mask}$ | $AR_{mask}$ | $AP_{box}$ | $AP^{50}_{box}$ | $AR_{box}$ |
> | ---------------- | -------- | ------------ | --------- | -------- | ----------- | --------- |
> | CutLER           | 4.19     | 8.42         | 10.86     | 4.94     | 9.61        | 13.69     |
> | CuVLER           | 2.92     | 6.16         | 7.32      | 3.51     | 7.47        | 13.15     |
> | **COLER (ours)** | **4.57** | **10.20**    | **13.75** | **5.98** | **12.44**   | **17.75** |
> | vs. SOTA (%)     | 9.07     | 21.14        | 26.61     | 21.05    | 29.45       | 29.66     |
>
> ### W1
>
> As mentioned in Section 4.3 (line 12), *we were unable to reproduce the official CuVLER results using the released code*; details are provided in Q1 below. From the results reported in CutS3D, CuVLER even performs worse than CutLER.
>
> *Pseudo-labels inevitably contain noisy samples, which is fundamentally different from supervised training*. Neither CutLER nor CutS3D evaluates the AP of pseudo-labels, and specialized loss functions are typically required to mitigate the impact of noisy labels. In addition, ***for existing methods such as CutLER and CutS3D, the hyperparameters used in pseudo-mask generation are selected based on trained model performance rather than the intrinsic quality of pseudo-labels.***
>
> Therefore, evaluating pseudo-label quality solely based on AP is insufficient, *while the main quantitative conclusions should be based on COLER.* **Furthermore, our training data only uses ImageNet, which is largely consistent with the setting in VoteCut.**
>
> *DiffNCut focuses on unsupervised object localization, which differs from our task of unsupervised instance segmentation.* The datasets and evaluation metrics are also different. Notably, recent works such as CutS3D (2025) and unMORE do not cite this paper.
>
> ### W2
>
> Density tuning is conceptually related to unsupervised graph clustering. ***However, for boundary enhancement and the combination of patch features with ranking, we are not aware of prior work adopting similar approaches. We kindly ask the reviewer to point out specific studies that utilize these two components.***
>
> **CuVLER, CutS3D, and unMORE all reuse the pipeline of CutLER.** In contrast, our method does not require a specialized loss function, and both training and self-training converge faster under the same experimental settings. This demonstrates that ***CutOnce generates highly effective pseudo masks for model training.***
>
> ### Q1
>
> We use ImageNet val (50k images) for pretraining and evaluate on COCO val2017 under the same experimental settings. None of the following methods use self-training. CuVLER* denotes the model trained by ourselves using the official code, while CuVLER uses the official pretrained weights.
> | Method | Pretrain | Specified loss | $AP_{mask}$ | $AP^{50}_{mask}$ |
> | ------------------ | -------- | -------------- | ------- | ------------ |
> | CuVLER (official)  | IN val   |     √       | 9.8     | 19.3         |
> | CuVLER* (official) | IN val   |     √    | 8.6     | 17.5         |
> | CuVLER             | IN val   |       ×     | 8.0     | 16.1         |
> | **COLER (ours) **      | IN val   |     ×    | **9.2**     | **19.2**        |
>
> ### Q2
>
> The OpenImages dataset contains a large number of objects, where COLER performs comparably to CutLER. ***Moreover, on Object365, which contains even more objects, our method still achieves the best performance.***
>
> It is also important to note that our pseudo-labels are at the patch level. We do not use CRF due to computational constraints, which also demonstrates that our method can achieve strong performance without CRF post-processing.
>
> ***We believe these results do not indicate insufficient generalization ability of COLER.*** On the contrary, the performance drop of CuVLER on KITTI raises more concerns regarding its generalization capability.

---

> > ### Author Rebuttal · Reviewer_NtDB · 2026-04-04
> >
> > I thank the authors for the detailed and thoughtful rebuttal.
> >
> > After reviewing the response, two points remain that I believe would further strengthen the paper:
> >
> > **1. Controlled experiment for the core claim.** The comparisons provided (CutS3D, Q1 table) are informative. However, since these comparisons involve differences in pipelines and loss functions alongside the label source, it would be more convincing to see a controlled experiment where the *same* detector is trained with CutOnce vs. VoteCut labels under *identical* settings (same loss, same data scale, same self-training rounds). Such an experiment would provide the most direct evidence for the paper's central contribution.
> >
> > **2. OpenImages ablation.** I was not able to find a direct response to my original Q2 regarding whether Boundary Augmentation contributes to the AP decline on OpenImages. While the Cityscapes results are helpful, an analysis on dense, multi-object benchmarks like OpenImages would provide additional valuable insight. It would be helpful if the authors could clarify this point.

---

> > > ### Author Response · Authors · 2026-04-05
> > >
> > > We thank you for your further feedback and for carefully reading our rebuttal. We are glad that you recognize the novelty and methodology of our work. Below, we provide additional clarification on the two questions you raised.
> > >
> > > 1. Based on the two follow-up works of CutLER, namely CuVLER and CutS3D, *we did not find the type of comparison you suggested*. ***To address your question, we provide such a comparison on COCO val2017 here.*** First, COLER shares the same detector with other methods. The loss functions are different (COLER does not use a specific loss, while other methods use their own designed losses). The datasets are partly the same (CuVLER uses ImageNet val, while others use ImageNet train). The number of self-training rounds is also partly the same (CuVLER does not use self-training, while others do, but the detailed settings are different). Results for CutLER are taken from the paper and our reproduction (which are consistent). CuVLER is reproduced by ourselves (the official version uses ImageNet val for training). CutS3D results are taken from the paper (no code is available and some data is missing). ***It is difficult to unify the loss functions: CutLER uses DropLoss, CuVLER uses Soft Target Loss, and CutS3D uses a modified Soft Target Loss, while our COLER does not use a specific loss.***
> > >
> > > | Method | Pretrain | Use CRF | Specific Loss | Train Rounds | Copy-paste | $AP^{mask}_{50}$ | $AP^{mask}$ |
> > > | ------ | -------- | ------- | ------------- | ------------ | ---------- | ---------------- | ----------- |
> > > | CutLER | IN train | √       | ×             | 1            | ×          | 15.8             | 7.7         |
> > > | CuVLER | IN train | √       | ×             | 1            | ×          | 15.1             | 7.3         |
> > > | CutS3D | IN train | √       | √             | 1            | ×          | -                | 8.5         |
> > > | COLER  | IN train | ×       | ×             | 1            | ×          | 19.2             | 8.2         |
> > >
> > > 2. In the rebuttal, we mentioned that this question may not be critical, but we still provide a clear answer here. As shown in the table below, boundary augmentation improves performance on OpenImages, although the gain is smaller compared to COCO and KITTI.
> > >
> > > | Method       | Boundary augment | Pretrain       | Train rounds | $AP^{mask}_{50}$ | $AP^{mask}$ |
> > > | ------------ | ---------------- | -------------- | ------------ | ---------------- | ----------- |
> > > | COLER (ours) | ×                | ImageNet train | 1+3          | 17.7             | 9.8         |
> > > | COLER (ours) | √                | ImageNet train | 1+3          | **18.2**             | **10.1**        |

---

### Official Review · Reviewer_49d1 · 2026-03-08

**Soundness:** 2
**Presentation:** 3
**Significance:** 2
**Originality:** 2
**Overall Recommendation:** 3
**Confidence:** 3

**Summary:**

This work is solving a challenging task in unsupervised instance segmentation and object detection. The proposed method can partition multiple instances in a single training image. Compared with the NCut algorithm in multi-object segmentation, the new method can do so without recursive steps.

**Compliance With Llm Reviewing Policy:**

Affirmed.

**Final Justification:**

Thanks for the detailed response. Some of my concerns have been resolved. Although there is some theoretical justification for certain modules, a comprehensive theoretical analysis of the entire method is still missing.

**Key Questions For Authors:**

Is there any mathematical model or theoretical framework to support the theoretical analysis?

**Limitations:**

The proposed method is heuristic and does not have a strong theoretical foundation.

**Strengths And Weaknesses:**

Strengths:

The task solving is challenging. Since this is no labels used for unsupervised object segmentation task. Particularly, when multi-object appeared in the training dataset.

No recurse steps are used for multi-object segmentation, compared with Ncut， which simplifies the segmentation pipeline and reduce computational complexity.

Weaknesses:

Since the main idea of the paper is more beyond NCut rather than a pipeline, it is better to add experiments comparing NCut with traditional NCut on the Berkeley segmentation dataset.

The experimental performance is worse than VoteCut (2024), and the improvements are minimal compared with the baseline. Since there is no 2025 paper compared, the “SOTA” claim is overclaimed. The experimental performance is a little bit weak.

The theoretical analysis is relately weak. There is no theory model or mathematical model for the theoretical foundation provided to show why the proposed method works. There is no clear reason why the proposed method is better than NCut. The proposed method is heuristic.

Other:

The presentation could be better. There is no method description in the introduction section. The text in Figure 1 is too small. It is better to reorganize Figure 1 to make the text bigger and clearer.

---

> ### Author Rebuttal · Authors · 2026-03-27
>
> We thank the reviewer for recognizing the difficulty of this work. The questions you raised are meaningful, and we will improve the clarity of our presentation.
>
> Our contributions are as follows: we introduce a new paradigm for multi-object segmentation using NCut, namely discovering multiple objects in a single pass without relying on clustering or recursion. **If our novelty is not convincing, please point out specific related work.**
>
> Weaknesses
>
> 1.The Berkeley Segmentation Dataset is relatively small and outdated, and has not been used in recent related work. We do not modify NCut itself, but instead optimize its input and output, with the goal of enabling the model to benefit from learning with pseudo-labels. Our method first proposes COLER, followed by CutOnce. COLER has stronger generalization ability. *In our response to reviewer NtDB, we showed the performance of COLER on complex scene-centric datasets such as Cityscapes, and we plan to include more datasets to further evaluate COLER.*
>
> 2.MaskCut shows a larger gap compared to VoteCut, while the gap after training becomes very close. From the CutS3D (2025 ICCV) paper, the model trained with VoteCut is inferior to that trained with MaskCut. *Combining these two observations, a higher AP of pseudo masks does not necessarily lead to better model performance under the same training settings.* Unlike supervised learning, pseudo masks inevitably contain erroneous samples, which are highly detrimental to training, even if correct samples dominate. Except for VoteCut, which provides quantitative analysis of pseudo masks, other related works such as MaskCut and LocalCut (the mask generator in CutS3D) do not report this part. *We do not avoid these counter-intuitive and unfavorable experimental results, in order to fully present the experimental process and make the results more credible.*
>
> *ImageNet val contains on average 1.6 objects per image.* ImageNet train does not have annotations, and the distribution is unlikely to differ significantly. In terms of the number of pseudo masks, our CutOnce is more reasonable, *while VoteCut generates an average of 8.9 per image,* which clearly contains many incorrect labels.
>
> Based on the above two points, this paper should focus on the results of COLER rather than CutOnce. If one focuses on CutOnce, then it should follow the same dataset configuration as TokenCut and compare the CorLoc metric.
>
> At the end of Section 4.1, we listed two works from 2025 but did not include them in the comparison. CutS3D does not provide code or pretrained weights, and does not report results on OpenImages. On the datasets it reports, our COLER outperforms it. unMORE uses both ImageNet and COCO, which is different from other methods such as CutLER that only use ImageNet; therefore, its evaluation on COCO is clearly not zero-shot, and comparing it with other methods is not fair.
>
> 3.***The first two modules have solid theoretical foundations and can be viewed as principled extensions of spectral clustering.*** Density tuning corresponds to adaptive kernel scaling for handling non-uniform feature density; boundary enhancement introduces high-frequency signal correction into low-frequency spectral embedding. The third module, patch features and object filtering, is a heuristic method designed to replace the heuristic used in TokenCut, which relies on the maximum feature value to localize a single object. *Similar to TokenCut, our CutOnce presents all engineering details in the main paper, whereas other methods are clearly more complex and do not disclose certain details.*
>
> The original text contains more intuitive reasoning than theoretical reasoning, which is a writing issue on our side. We will include theoretical derivations in the main text and move intuitive explanations to the appendix. *Since the theory of spectral clustering is relatively old and difficult to follow, we did not include extensive derivations in the main text.* ***Due to space limitations, we provide detailed theoretical derivations for these two modules and demonstrate their validity at the anonymous link: https://anonymous.4open.science/r/icml_2026_rebuttal-20F8/README.pdf.***
>
> 4.We accept the reviewer’s comments on the introduction section.
>
> Limitations
> See the theoretical derivation in Q3.

---

> > ### Author Rebuttal · Reviewer_49d1 · 2026-04-04
> >
> > Thanks for the response, but some of my concerns remain unresolved. In particular, the theoretical analysis is still weak, and there is no clear theoretical justification supporting the effectiveness of the method.

---

> > > ### Author Response · Authors · 2026-04-06
> > >
> > > Thank you for partially recognizing our rebuttal and for not raising concerns about the novelty of our work. We would like to clarify that the density tuning module and the boundary enhancement module are applications of well-established ideas from spectral clustering theory, while the feature and filtering components are heuristic. The last paragraph of each module in Section 3.2 of the paper explains these points in detail. In our previous rebuttal, we also provided theoretical derivations for the two modules that have theoretical foundations.

---

### Official Review · Reviewer_hCyK · 2026-03-12

**Soundness:** 3
**Presentation:** 3
**Significance:** 3
**Originality:** 3
**Overall Recommendation:** 4
**Confidence:** 3

**Summary:**

This paper studies unsupervised instance segmentation and object detection. It follows the common 2-stage pipeline: first generate pseudo masks by object discovery, then train a class-agnostic detector with self-training. The main method is CutOnce, which uses one-time NCut with density-tuned similarity, boundary augmentation, and a ranking filter to produce multiple object masks. Based on this, COLER is trained with pseudo labels and achieves better results than recent methods on several benchmarks.

**Compliance With Llm Reviewing Policy:**

Affirmed.

**Final Justification:**

I thank the authors for the careful rebuttal and the clear replies to my questions. The response is useful to clarify several points, especially the practical setting of the graph threshold, the reason for using representative-class update with neighbor propagation, and the possible failure cases of the method. These explanations make the paper easier to understand and help me see the design logic more clearly. At the same time, I still have some reservation about the practical robustness of the graph construction and whether the full pipeline is fully necessary in its current form. So my main concerns are only partially addressed, and my score stays the same.

**Key Questions For Authors:**

- Can the authors clarify more clearly which parts of the improvement come from better pseudo masks, and which parts may come from the detector training recipe?

- Can the authors explain more clearly the exact difference between CutOnce and CutOnce+ in the whole pipeline, and why the paper finally chooses the current setting?

- Can the authors comment more on the main failure cases of the method, especially for crowded or overlapping objects?

**Limitations:**

The method still seems to have difficulty in some complex scenes, especially when objects are crowded or heavily overlapped. Also, the final performance still depends on the overall training pipeline, so the paper should be more careful in explaining this point.

**Strengths And Weaknesses:**

Strengths:

The paper studies an important problem. The proposed mask generation part is relatively simple and practical, and does not need extra training. The experimental section is also quite broad, covering several datasets and including ablation.

Major issue:

- The comparison setting is not fully clear to me. In some places, it is hard to judge whether the gain mainly comes from the pseudo mask quality or from other training details, such as data split, self-training setting, or some implementation choices. I think the paper should explain these settings more clearly and directly.

- The paper presentation still needs improvement. Some writing is not very smooth, and a few claims feel a bit too strong. Also the formatting looks not very clean in some parts. This affects readability and also confidence in the paper.

Minor issue:
- The discussion about limitations is still not deep enough. For example, crowded scenes, heavy overlap, and coarse boundaries seem to be difficult cases, but the paper does not analyze them very clearly.

- The role of some design choices is not always explained well enough. For example, the relation between CutOnce and CutOnce+ is a little confusing when reading the full paper, and the practical reason behind some choices can be better clarified.

---

> ### Author Rebuttal · Authors · 2026-03-27
>
> We sincerely thank the reviewer for the recognition of our work. We will do our best to address your concerns, and we would greatly appreciate any further valuable suggestions you may have.
>
> **Main Issues**
>
> 1. The performance improvement indeed comes from CutOnce. Our training setup is made as consistent as possible with CutLER and CuVLER, and these details are described in the paper. Subsequent methods after CutLER, such as CuVLER and CutS3D, mainly improve the pseudo-mask generation stage while largely retaining the training pipeline of CutLER to demonstrate the effectiveness of their modifications. There is no issue with dataset splits, as these datasets are predefined and do not introduce discrepancies. Our self-training setting is also nearly identical to CutLER. In the ablation study, we present results before self-training on two datasets, and we summarize here that our method achieves the best performance. ***Similar to TokenCut, we present all engineering details in the main paper, whereas other methods are more complex and do not clearly disclose such details.*** *We kindly ask the reviewer to explicitly point out which parts of our implementation are not clearly described in the paper.*
>
> 2. We will incorporate feedback from all reviewers to improve the clarity and fluency of the paper. *If you could indicate specific paragraphs, we would be very grateful.*
>
> **Minor Issues**
>
> 1. At the current stage of research, even in supervised settings, crowded scenes and severe occlusions remain challenging problems. Our understanding of these issues is still limited, and we appreciate your understanding. Based on our limited experience, handling occlusions may involve introducing occlusion labels, but this inevitably increases annotation cost.
>
> 2. As mentioned in the original paper, the only difference between CutOnce and CutOnce+ is whether CRF post-processing is applied. *Due to limited CPU and GPU resources, we initially decided not to use CRF. However, we observed that CutOnce underperforms MaskCut on certain metrics on COCO, which motivated us to introduce CutOnce+.*
> This design demonstrates that:
> 1) our CutOnce does not rely on post-processing;
> 2) when CRF is applied, CutOnce+ can further improve performance, meaning our COLER can also benefit from it.
>
> ***Since applying CRF on ImageNet train (1.28M images) is extremely time-consuming (especially for repeated ablation studies)***, we instead conduct CRF experiments on ImageNet val (50K images) and evaluate on COCO val2017. As shown in the table below, CRF has a positive effect on our method.
>
> | Method | CRF | Pretrain | Train Rounds | AP$_{50}^{mask}$ | AP$^{mask}$ |
> |--------|-----|----------|--------------|------------|---------|
> | COLER  | ×   | IN val   | 1            | 18.9       | 8.8     |
> | COLER+ | √   | IN val   | 1            | 19.6       | 9.2     |
>
> **Key Issues**
>
> 1. ***Most improvements are made in the pseudo-mask generation stage, while the training stage removes the need for specialized loss functions.***
>
> 2. **As mentioned earlier, we do not use CRF mainly due to hardware limitations that prevent us from completing large-scale experiments. The use of CRF on smaller datasets is intended to demonstrate that CRF can further enhance our method.**
>
> 3. The upper bound of our CutOnce and related methods such as MaskCut and LocalCut (the mask generator in CutS3D) is determined by the self-supervised model DINO, while the lower bound is constrained by the NCut algorithm applied to DINO features. Even supervised and self-supervised models struggle with these challenging scenarios, let alone unsupervised methods. *The motivation of this work is to surpass methods like CutLER and CuVLER across multiple datasets, rather than specifically targeting these extremely difficult cases.* Currently, we observe that COLER shows significant advantages on complex scene-centric datasets such as Cityscapes (see our response to reviewer NtDB for details), although the absolute performance is still relatively low.

---

> > ### Author Rebuttal · Reviewer_hCyK · 2026-04-04
> >
> > The rebuttal addressed part of my concerns, but my main concern is still not fully resolved. Therefore, I maintain my score.

---

### Official Review · Reviewer_2aKb · 2026-03-13

**Soundness:** 3
**Presentation:** 3
**Significance:** 2
**Originality:** 2
**Overall Recommendation:** 3
**Confidence:** 5

**Summary:**

This paper proposes COLER for unsupervised instance segmentation and object detection. The framework relies on CutOnce to generate pseudo masks and then trains a detector using these masks. CutOnce is based on the Normalized Cut (NCut) formulation. First, an affinity matrix is constructed from the cosine similarity between DINO patch features. A local density tuning step is then applied to adjust the affinities, after which NCut is performed to obtain an eigenvector representing an attention map. To smooth this eigenvector, the method subtracts a boundary component computed from local differences. The resulting vector is thresholded to separate foreground and background. To obtain multiple objects, an additional thresholding step based on the ratio of summed feature responses is applied, producing the final object masks. The detector is then trained with these pseudo labels, followed by several rounds (about three) of self-training. The proposed method is evaluated against two baselines on several benchmarks. While the generated pseudo masks themselves show relatively modest performance, the final trained detector achieves strong results.

**Compliance With Llm Reviewing Policy:**

Affirmed.

**Final Justification:**

I thank the authors for their detailed responses and efforts to improve the paper. While I believe the work has potential, in its current form it falls short of the bar for a top conference. I recommend strengthening intuitive, clearly interpretable empirical comparisons along with more thorough analysis.

**Key Questions For Authors:**

See weaknesses

**Limitations:**

Yes

**Strengths And Weaknesses:**

# Strengths
### Overall method make sense

- The proposed components are generally reasonable and well motivated.
- In particular, both the density tuning module and the boundary augmentation step aim to smooth the attention map, which could plausibly help in scenarios with multiple objects.

### The paper is easy to follow

- The paper is easy to follow overall. The method is intuitive and the figures and examples help clarify the pipeline.

### Experiment result is intersting

- It is interesting that the pseudo masks themselves are not particularly strong, yet the final trained model achieves competitive performance.
- The inclusion of hyperparameter analysis and ablation studies is helpful in understanding the behavior of the method.


# Weaknesses
### The experimental results are not fully convincing in several aspects.

- Although the final model performance is strong, the evaluation metrics that directly measure pseudo mask quality show relatively weak performance. Additional analysis would be helpful to explain this gap.
- In particular, it would be useful to demonstrate more clearly in what situations CutOnce produces better pseudo labels. Further analysis of datasets where the performance is relatively low (e.g., in Table 4) would also be informative.
- In Table 3, the comparison between CuVLER and COLER (IN train) is somewhat unclear, and it would be helpful to clarify whether the configurations are fully comparable.

### The choice of baselines is somewhat limited

- Only two baselines are included, and it is unclear whether they adequately represent the current state of the art. It would strengthen the evaluation if the method were compared with additional recent approaches.
- For example, reproducing baselines such as unMMORE, or providing comparisons on datasets such as VOC, KITTI, OpenImages, or Objects365, would make the evaluation more complete.

### The method appears relatively sensitive to hyperparameters

- While the paper includes a sensitivity analysis, the results suggest that performance drops for most settings other than the chosen configuration, often falling below previous methods.

### proposed ideas do not appear substantially novel

- Many components resemble incremental modifications to earlier work such as TokenCut.
- Techniques such as smoothing attention maps, removing boundary components through filtering, or applying thresholding strategies have been explored previously,
- The proposed design can largely be viewed as incremental refinements rather than fundamentally new ideas.

---

> ### Author Rebuttal · Authors · 2026-03-28
>
> We thank the reviewer for carefully reading our paper. We are happy to address your concerns. The description of the method section has been revised. To further enhance your understanding of our approach, we provide detailed theoretical derivations at the anonymous link: https://anonymous.4open.science/r/icml_2026_rebuttal-20F8/README.pdf
>
> ***Our CutOnce opens up a third direction for using NCut to segment multiple instances, i.e., without relying on clustering or recursion. If you have concerns regarding our claimed novelty, please point out specific related work.***
>
> **Weaknesses**
>
> 1. **AP can evaluate the quality of pseudo masks, but a higher AP of pseudo masks does not necessarily imply better model performance under the same training setup.** Unlike supervised learning, erroneous samples in pseudo labels negatively affect model training. *Our training pipeline does not require designing specific loss functions to handle noisy pseudo labels, which demonstrates the robustness of our method.* ***ImageNet Val contains on average 1.6 objects per image, while VoteCut detects 8.9 objects, clearly introducing many incorrect labels.*** In contrast, MaskCut and our CutOnce detect 1.9 and 1.8 objects respectively, which are closer to the true distribution. In terms of AP, VoteCut significantly outperforms MaskCut, yet the final performance of CuVLER and CutLER is very similar. ***From the CutS3D paper, CuVLER even performs worse than CutLER.*** Moreover, we are unable to reproduce the results reported in the CuVLER paper using their official code. In fact, except for VoteCut, other methods such as CutLER and CutS3D do not evaluate pseudo-mask AP. *Their ablation studies (including ours) select hyperparameters based on the performance of the trained models.* ***We explicitly report this counterintuitive pseudo-mask evaluation result for completeness.***
>
> 2. *Our performance on OpenImages is not the best, but compared to the degradation of CuVLER on KITTI, we consider this limitation acceptable.* Moreover, we achieve strong performance on denser datasets such as Objects365 and LVIS. *We believe that generating pseudo labels on ImageNet train with CRF would further improve results.* On complex scene-centric datasets such as Cityscapes, our COLER shows significant advantages (***see our response to reviewer NtDB for details***).
>
> 3. Regarding training data, the official CuVLER configuration uses ImageNet val and does not report results on ImageNet train. We use ImageNet train for COLER and ImageNet val for COLER* to **demonstrate the effectiveness of CutOnce under different data scales**.
>
> 4. Regarding baseline selection, the last paragraph of Section 4.1 clearly explains why CutS3D (ICCV 2025) and unMORE (ICML 2025) are not included. *If there are other relevant baselines, we kindly ask the reviewer to point them out.*
>
> 5. We believe that ***unMORE’s official setting, which uses both ImageNet and COCO, is unfair compared to methods that only use ImageNet, especially since COCO is a key evaluation benchmark.*** We evaluated unMORE trained only on ImageNet, and the results on COCO val2017 are shown below. *If you disagree with this setting, please clarify the reasons.*
>
> | Method | Pretrain | Train rounds | AP$^{mask}_{50}$ | AP$^{mask}$ |
> |--------|:--------:|:------------:|:------------------------------:|:----------------:|
> | CutLER | ImageNet train | 1 | 17.7 | 8.8 |
> | unMORE | ImageNet train | 1 | 19.3 | 9.2 |
> | COLER (ours) | ImageNet train | 1 | 20.6 | 9.6 |
>
> 6. $\tau$ has a critical threshold; exceeding this value introduces many very small regions.
> $k$ denotes the number of neighbors. Since features are relatively smooth in most regions, the choice of $k$ has limited impact on the final results.
> **$T_0$ and $\alpha$ directly control the scaling of all elements in the matrix $\mathbf{W}$, which significantly affects the final results.** These hyperparameters should be chosen more conservatively.
>
> 7. ***We acknowledge that our method is developed through incremental modifications, but it can segment multiple objects with an unfixed number. Currently, no existing method achieves this. If such work exists, please point it out.*** As shown in Figure 2, our approach is a replacement-based improvement, while others are incremental.
>
> 8.  ***First, our second module does not remove boundaries via filtering; instead, it uses local gradients to make feature distributions more uniform, allowing more objects to be “exposed”.*** We kindly ask you to provide specific references for the techniques you mentioned, *as we have not yet found similar work.*
>
> 9. ***As stated at the beginning, our method is not fundamental but is pioneering.*** If there exists work that segments multiple objects with NCut without using clustering or recursion, ***please point it out.*** We believe that CutOnce goes ***beyond the two paradigms (clustering and recursion) proposed in the original NCut work*** over 20 years ago.

---

> > ### Author Rebuttal · Reviewer_2aKb · 2026-04-04
> >
> > Thank you for the additional experiments and responses. I appreciate the effort to further clarify the paper and I'm happy to see the work improving. Below are my remaining questions and concerns.
> >
> > ## Gap between pseudo label performance and final model performance
> >
> > I agree that higher AP of pseudo masks does not necessarily imply better downstream performance. However, some level of correlation is generally expected. What I am asking for is a more analytical justification that the pseudo labels from CutOnce are indeed better for training. If the claim is that the proposed pseudo labels contain less noise, this should be supported by more targeted analysis. For example:
> >
> > - Evaluating precision-oriented metrics
> > - Analyzing the alignment between the predicted number of objects and the ground-truth object statistics across datasets
> >
> > For instance, on COCO val2017, CutOnce produces approximately 1.8 masks per image. It is unclear whether this reflects improved precision or simply severe under-detection, and how this trade-off affects training.
> >
> > ## Performance on OpenImages
> >
> > It is unclear how the degradation of CuVLER on KITTI justifies the relatively low performance of COLER on OpenImages. Could you provide a more detailed analysis of the failure cases on OpenImages?
> >
> > If the absence of CRF is a key factor, it would be helpful to evaluate the performance when applying CRF to the pseudo labels.
> >
> > ## Training configuration on Table 3
> >
> > The “vs. SOTA (%)” comparison in Table 3 appears to be based on configurations that are not fully aligned (e.g., different training data).This makes the comparison potentially misleading, and the current presentation may overstate the improvement. I recommend revising this comparison to ensure fairness and clarity.
> >
> > ## Comparison with the baseline
> >
> > Thank you for including additional experiments with unMORE. While I do not strongly disagree with the current setting, it would be more convincing to also evaluate COLER under the original training setup of unMORE (e.g., including COCO), so that the reported numbers can be directly compared with prior paper.
> >
> > Additionally, I would like to emphasize that the difficulty of reproduction or lack of public code does not fully justify limited baseline coverage. Even reporting the numbers from re-implemented codebase (even if its not exactly matches with previous paper) would help strengthen the evaluation.
> >
> > ## Hyperparameter sensitivity
> >
> > If the method is sensitive to hyperparameters, it would be helpful to provide practical guidance on how to select them. For example: Is it possible to select them using a small labeled validation set? Providing such guidance would improve the usability of the method.

---

> > > ### Author Response · Authors · 2026-04-06
> > >
> > > Thank you for carefully reading our rebuttal. We will now respond to your remaining questions in detail.
> > >
> > > ### Gap between pseudo label performance and final model performance
> > >
> > > First, as stated in the abstract, the goal of this paper is to make COLER achieve the best performance, rather than making CutOnce itself the best. In related works such as CutLER, CuVLER, CutS3D, and unMORE, pseudo masks are not evaluated. VoteCut and our method use the same evaluation protocol. We do not agree with the evaluation method you suggested; we believe that comparing the performance of models trained with pseudo labels is more meaningful. If pseudo labels are to be evaluated separately, the experimental setup should follow that of TokenCut, which is not aligned with the goal of this paper.
> > >
> > > Missed detections on COCO val2017 do not affect model training, since we only use ImageNet for training. We do not consider training on COCO for two main reasons: (1) COCO has mask annotations while ImageNet does not, so it cannot demonstrate the unsupervised reality; (2) our model is zero-shot, and COCO is an important evaluation dataset.
> > >
> > > We should focus more on the final impact of pseudo labels on model performance rather than the quality of pseudo labels themselves. From the results in the CutS3D paper, CuVLER performs worse than CutLER, which suggests that false detection are more problematic than missed detection. The CuVLER paper also shows that using a self-supervised model alone significantly reduces performance. This means that VoteCut largely improves performance by combining multiple self-supervised models.
> > >
> > > This part reflects improvement in precision. The paper mentions the impact of confidence on AP evaluation. In principle, the confidence of pseudo labels in VoteCut ranges from 0 to 100, while for MaskCut and CutOnce it ranges from 50 to 100 (we assign confidence score, otherwise AP cannot be computed). Therefore, the improvement of CutOnce over MaskCut is clear, while the improvement of VoteCut over MaskCut is uncertain.
> > >
> > > ### OpenImages performance
> > >
> > > In the rebuttal, we stated that our performance on OpenImages is not unacceptably worse than previous methods. In fact, our result lies between CutLER and CuVLER.
> > >
> > > The example of CuVLER on KITTI is used to show that if CuVLER is not considered problematic, then our COLER should not be overly questioned for similar behavior. CutS3D does not report results on OpenImages, which suggests that this dataset may not be a major focus.
> > >
> > > Providing detailed analysis of failure cases on OpenImages may not be very meaningful. The goal of this paper is to make COLER outperform other related methods, rather than to analyze failure cases.
> > >
> > > As mentioned in the paper, CRF is time-consuming, and our hardware does not support running all experiments with CRF. Achieving strong performance without CRF further shows the generality and efficiency of our method.
> > >
> > > ### Training configuration in Table 3
> > >
> > > The official CuVLER model is trained on ImageNet val, while we can also use ImageNet train. The paper clearly states that training CuVLER on ImageNet train leads to worse performance than the official model. Therefore, we choose to use the official model weights with better performance.
> > >
> > > Regarding the concern about overstating the improvement, we kindly ask you to provide more specific reasons and possible solutions.
> > >
> > > ### Comparison with baselines
> > >
> > > We do not agree with this point. As mentioned in the paper and the rebuttal, our model is a zero-shot model, while the original training setting of unMORE is not zero-shot. Therefore, unMORE can only be included in the comparison under a zero-shot setting.
> > >
> > > We agree with your point. As shown in the rebuttal, our COLER outperforms unMORE. In our response to reviewer NtDB’s rebuttal comments, our COLER also outperforms CutS3D.
> > >
> > > ### Hyperparameter sensitivity
> > >
> > > In the rebuttal, we stated that the ranges of $T_0$ and $\alpha$ are large, rather than being sensitive as you suggested. The difference in $AP^{mask}_{50}$ is larger than in $AP^{mask}$. The purpose in the paper is to show which parameter settings have larger impact on performance. When we choose similar parameter values, the impact on performance is small.
> > >
> > > | $T_0$            | 0.9  | 1.0  | 1.1  | $\alpha$         | 0.4  | 0.5  | 0.6  |
> > > | ---------------- | ---- | ---- | ---- | ---------------- | ---- | ---- | ---- |
> > > | $AP^{mask}_{50}$ | 19.2 | 19.6 | 19.3 | $AP^{mask}_{50}$ | 19.3 | 19.6 | 19.4 |

---

### Decision · Program_Chairs · 2026-04-30

**Decision:**

Reject

**Comment:**

This paper presents a simple yet effective pseudo-labeling pipeline for unsupervised instance segmentation and object detection. It has been evaluated by expert reviewers in relevant fields. The reviewers recognized that the target task is important and challenging, the motivation is sound and reasonable, and the proposed method is simple and effective. However, they also raised multiple crucial concerns with
1. insufficient evaluations and in-depth analysis (All),
2. incremental novelty (2aKb, NtDB),
3. lack of justification for limited quality of pseudo labels (2aKb, hCyK),
4. limited performance gain (49d1, NtDB),
5. lack of theoretical analysis (49d1),
6. sensitivity to hyper-parameters (2aKb), and
7. weak quality of presentation (hCyK, 49d1).

The authors responded to these comments through the rebuttal and subsequent responses, but failed to fully assuage some of them. In the post-rebuttal phase, the negative reviewers still concerned with the insufficient evaluation, lack of analysis on the quality of pseudo labels, hyper-parameter sensitivity, lack of theoretical justification for the entire method, and marginal performance gain. Even the only positive reviewer had some concerns that are not fully assuaged such as those with potentially limited practical robustness and lack of justification for the proposed method. In the end, the majority of the reviewers kept their negative ratings, and the only positive reviewer did not strongly champion this paper. Putting these together, the AC considers that the remaining concerns outweigh the positive comments and the rebuttal, and thus regrets to recommend rejection. The authors are encouraged to improve the paper with the valuable comments from the reviewers, and resubmit to next venues.